# Biofilm Formation by Mutant Strains of Bacilli under Different Stress Conditions

**DOI:** 10.3390/microorganisms11061486

**Published:** 2023-06-02

**Authors:** Margarita Sharipova, Natalia Rudakova, Ayslu Mardanova, Vladimir Evtugyn, Yaw Akosah, Iuliia Danilova, Aliya Suleimanova

**Affiliations:** 1Institute of Fundamental Medicine, Kazan Federal University, Kremlevskaya St. 18, 420008 Kazan, Russia; mardanovaayslu@mail.ru (A.M.);; 2Interdisciplinary Center of Analytical Microscopy, Kazan Federal University, Paris Commune St. 9, 420008 Kazan, Russia; 3Department of Molecular Pathology, NYU College of Dentistry, 345 E. 24th Street, New York, NY 10010, USA

**Keywords:** *B. subtilis*, protease-deficient strain, regulatory mutant, metal ions, salt stress, confocal microscopy

## Abstract

*Bacillus subtilis* is traditionally classified as a PGPR that colonizes plant roots through biofilm formation. The current study focused on investigating the influence of various factors on bacilli biofilm formation. In the course of the study, the levels of biofilm formation by the model strain *B. subtilis* WT 168 and on its basis created regulatory mutants, as well as strains of bacilli with deleted extracellular proteases under conditions of changes in temperature, pH, salt and oxidative stress and presence of divalent metals ions. *B. subtilis* 168 forms halotolerant and oxidative stress-resistant biofilms at a temperature range of 22 °C–45 °C and a pH range of 6–8.5. The presence of Ca^2+^, Mn^2+^ and Mg^2+^ upsurges the biofilm development while an inhibition with Zn^2+^. Biofilm formation level was higher in protease-deficient strains. Relative to the wild-type strain, *degU* mutants showed a decrease in biofilm formation, *abrB* mutants formed biofilms more efficiently. *spo0A* mutants showed a plummeted film formation for the first 36 h, followed by a surge after. The effect of metal ions and NaCl on the mutant biofilms formation is described. Confocal microscopy indicated that *B. subtilis* mutants and protease-deficient strains differ in matrix structure. The highest content of amyloid-like proteins in mutant biofilms was registered for *degU*-mutants and protease-deficient strains.

## 1. Introduction

The rhizosphere is a complex ecosystem that includes plant roots and a layer of soil adjacent to the roots with a high content of microorganisms, which is the site of interaction between plants and bacteria. The interaction occurs as a result of root colonization by plant growth-promoting rhizobacteria (PGPR), among which representatives of the *Bacillaceae* family are deemed highly significant. The stages of colonization include chemotaxis, adhesion, aggregation, and biofilm formation which are largely determined by biological and environmental factors [1]. Colonization and formation of interactions of bacteria with plant roots activate chemotaxis [2,3]. Interaction in the rhizosphere occurs through a complex set of signaling molecules that control the behavior of microorganisms in communities. Specific gene products, among which transcription factors play a key role mediate the formation and development of the rhizomicrobiome [4].

*Bacillus* sp. are Gram-positive PGPRs useful for agriculturally important plants. *Bacillus subtilis* is a model organism for the study of biofilms, and the molecular networks and regulatory systems for biofilm formation in these bacteria have been well characterized. The transcription factors Spo0A, DegU, and AbrB are involved in the control of biofilm formation; they regulate the expression of genes associated with the production of the cell matrix [5,6,7]. Several studies have confirmed root exudates and their individual components (sugars, organic acids, amino acids) as important promoters of the growth and biofilm formation of PGPR [8,9,10]. Nevertheless, the influence of environmental and stress factors on these processes remains poorly understood. Understanding the process of colonization by *Bacillus* strains and the search for factors influencing this process is necessary to control the use of rhizobacteria in agriculture [11]. An important role in the process of colonization is played by the secreted enzymes of bacilli, which are involved in the transformation of proteins to peptides, many of which are important signaling molecules in the rhizosphere [12]. In addition to physical and chemical factors, bacteria united by a biofilm develop and differentiate under the action of enzymes. Information on the control of enzymes in the biofilm development cycle is insufficient. Such knowledge is useful for the development of a general strategy for the management of biofilms of spore-forming *B. subtilis* bacteria, which is considered an effective biocontrol tool.

This study aimed to investigate the biological and genetic factors influencing the biofilm formation of the potential plant root colonizer *B. subtilis* (strain *B. subtilis* 168). The results obtained from this study are essential for determining the optimal conditions for biocontrol activity to maximize crop productivity.

## 2. Materials and Methods

The strains used in the study are shown in the Table 1.

Cultivation of the strains was performed on a synthetic-E medium, of the following composition (g/L): L-glutamic acid—20; citric acid—12; glycerol—80; NH_4_Cl—7; MgSO_4_*7H_2_O—0.5; K_2_HPO_4_—0.5; CaCl_2_*2H_2_O—0.15; FeCl_3_*6H_2_O—0.04, MnSO_4_*H_2_O—0.148, pH 7.4, T = 37 °C [13]. Seed served 16-h inoculum (1% *v*/*v*). Bacterial growth was monitored by the change in the optical density of the culture at 600 nm (xMark spectrophotometer BioRad, Hercules, CA, USA). Biomass was expressed in absorbance units. Spore formation was determined by counting cells and spores by Peshkov microscopy method mode microscope Carl Zeiss Jena (Germany) in 1600 times magnification in four visual fields. The number of free spores was expressed as a percentage of the total number of vegetative and sporulating cells. Solutions of antibiotics (kanamycin and tetracycline) were prepared separately and added to the medium immediately before inoculation of the culture at a final concentration of 10 μg/mL. NaCl was added to the medium in the dry form before autoclaving to a final concentration in the medium of 1 M. Hydrogen peroxide (H_2_O_2_) was added sterile for 24 h of biofilm at final concentrations of 1, 5, and 10 mM. Metal ions in the form of salts of CaCl_2_, MgCl_2_, MnCl_2_, and ZnCl_2_ were sterilized separately and introduced into the medium in final concentrations of 1, 3, and 9 mM before inoculation. The assessment of the influence of all stress factors (NaCl, H_2_O_2_, metal ions) was carried out on an E-medium, pH 7.4 at 37 °C. Biofilm formation is defined by the method set incubation with crystal violet (CV) [14] with modification [15]. The cell culture with optical density OD600 = 0.1 was introduced into 96-well plates with a U-shaped bottom, 100 µL per well, and incubated at 37 °C under a lid. After incubation, the culture liquid with the plankton culture was removed from the plates, the plate was washed with water and dried at room temperature. 125 μL of 0.1% crystal violet dye (Sigma-Aldrich, St. Louis, MO, USA) was added to the wells and kept for 10 min at room temperature. The plates were then washed again with water and dried at room temperature. 200 μL of dimethyl sulfoxide (DMSO) (Sigma-Aldrich, USA) was added to each well and kept for 15 min at room temperature, during which the crystal violet bound to the biofilm structure dissociates into a DMSO solution. The contents of the wells were taken (125 μL), and transferred to the wells of a flat-bottomed 96-well plate for measurement on an xMark spectrophotometer (BioRad, USA) at a wavelength of 570 nm. Sterile E-medium without cell culture was used as a control for the biofilms formation during the experiment. The presence of the ability to synthesize amyloids in the studied strain was established by testing using a medium containing 25 μg/mL of Congo Red dye [16]. The culture plates were incubated at a temperature of 37 °C for 12–48 h and the binding of the Congo Red dye by the cells was noted by changing the color of the colonies.

The relative content of amyloid fibrils in the bacterium *B. subtilis* was estimated according to the method described in the article by Gophna et al., 2001 [17]. To study the relative content of amyloid fibrils, *B. subtilis* bacteria were grown on plates with agar medium E for 2 days at 37 °C. Colonies were scraped off and resuspended in a sterile 0.9% NaCl solution. The concentration of bacteria was determined by measuring the optical density at 570 nm. The cell suspension was diluted to a turbidity index OD570 = 0.9, and 1 mL was taken and centrifuged for 10 min at 14,000 rpm. After removing the supernatant, bacteria were resuspended in 1 mL of Congo Red solution (0.002% in 0.9% NaCl) and incubated for 10 min at room temperature. Next, centrifugation was repeated under the same conditions, the supernatant was taken, and the optical density was measured at 500 nm. The intensity of binding of the Congo Red dye by the cells was assessed by the decrease in optical density relative to the initial solution of the Congo Red dye. To prepare biofilms for SEM, a 16-h cell culture grown at 37 °C was used. Bacteria were grown on PVC catheters placed in the wells of a plastic 12-well cell culture plate (Corning Costar, Corning, NY, USA). Catheters were cut out 1 × 1 cm^2^ in size and sterilized at 1 atm., 121 °C within 30 min. The required volume of cell culture was added to each well so that the final optical density (OD600) was 0.2. The plates were incubated for 12 h to 2 days at 37 °C. The catheters were washed with phosphate buffer PBS and the biofilm samples were chemically fixed in 1.5% glutaraldehyde for 12–16 h. Next, the samples were dehydrated in a series of alcohols with increasing concentration—30%, 40%, 50%, 70%, 80%, and 90% for 15 min each.

All analyses were performed at least on four biological replicates. The obtained data were processed using Statgraphics Plus 5.0. and GraphPad Prism 7.05 statistical software, and are presented as the mean ± standard deviation (SD). Student’s *t*-test analysis was used to calculate the data variance with *p* < 0.05 (*) and *p* < 0.01 (**) representing a significant difference.

## 3. Results

Analysis of the growth dynamics, biofilm formation, and sporulation of the *B. subtilis* strain showed that it develops biofilm beginning at 12 h of bacterial growth and reaches a maximum at 48 h in the stationary phase of culture growth. Biofilm dispersion, which began after 70 h correlated with the cell culture dying phase. Spores appeared in the medium at 46 h during the phase of biofilm development, and active sporulation occurred after 60 h at the stage of biofilm dispersion. Thus, it was observed that the processes of biofilm formation and sporulation, are separate and do not occur simultaneously, but develop sequentially (Figure 1a).

Based on the ultrastructural data, the *B. subtilis* 168 biofilm is formed in 2 days (Figure 1b). At 24 h of cultivation, cells begin to adhere to the surface. By the 36th h, the cells produce extracellular polymeric compounds and form clusters, a matrix appears that binds culture cells together, and at 48 h a mature biofilm with bacterial cells immersed in a dense protective matrix is observed.

The organization of the *B. subtilis* biofilm can begin at 10 °C, but the structure is well favored and maintained at a temperature range from 22 °C to 45 °C. At 50 °C the process occurs at a very lagged rate (Figure 2a). The bacilli are capable of forming biofilms in a slightly acidic environment (pH 5–pH 6.0). However, the optimal pH for biofilm formation ranges from 7.4–pH 8, a subsequent increase in pH to 8.5–9.0 plummets the level of biofilm formation (Figure 2b). The optimal pH for biofilm formation is closer to neutral. Under natural conditions, such a pH range is essential for the active colonization of plant roots with bacilli biofilms, which in turn facilitates the formation of mutually beneficial symbiotic relationships [18]. The upsurge in studies of this kind is relevant and promising for the advancement of agrobiotechnology since soil strains of *B. subtilis* are considered effective biocontrol agents in plant protection against soil pathogens [19,20,21,22].

The effectiveness of colonization control to a large extent depends on the content of metals in the rhizosphere, therefore, deciphering the role of metal ions in the formation of bacilli biofilms is imperative [23]. We studied the effect of divalent metal ions on the development of *B. subtilis* 168 biofilms (Figure 2).

The introduction of Ca^2+^ ions at a concentration of 3 mM or lower led to an increase in the level of *B. subtilis* 168 biofilm formation by up to 45% at 12 h of growth and up to 11% at 36 h of growth. With an increase in the concentration of exogenous calcium to 9 mM, the film formation was partially inhibited by 10%. Interestingly, it was noted that calcium effectively acts precisely at the early stage of *B. subtilis* biofilm maturation (Figure 2). There is evidence that Ca^2+^ ions play an important role in the adhesion of bacilli cells during biofilm formation [24]. The presence of calcium in the medium affects the regulatory pathways of *B. subtilis* 168 during flagella development and the synthesis of biofilm matrix components [25]. While analyzing the transcriptome of *B. subtilis* 168, the authors identified 305 genes, the expression of which is affected by calcium, including genes for the biofilm matrix biosynthesis and genes encoding the biosynthesis of spore coat polysaccharides. [25]. The interactions between the microorganisms and Ca^2+^ enhance both the extent and the rate of biofilm development with increasing Ca^2+^ concentration [26]. The authors note that the Ca^2+^ concentration also correlates with the *B. subtilis* biofilm morphology. Recent studies have shown that microbial biofilms contain an organized internal mineral structure composed of crystalline calcium carbonate (calcite), which largely determines their three-dimensional morphology [27].

Similar data were obtained when Mn^2+^ ions were introduced into the medium. The results showed that Mn^2+^ is essential at the starting point of *B. subtilis* 168 biofilm maturation; an increase in the level of film formation was observed up to the 36th h at Mn^2+^ concentrations of 1–3 mM (Figure 2). The absence of inhibition following the treatment of cells with Ca^2+^ and Mn^2+^ indicates their non-toxicity to biofilm formation. There is evidence that the exogenous presence of Mn^2+^ ions promotes the formation of stable biofilms in bacilli by participating in the activation of phosphorel, which is necessary for phosphorylation of the global transcription factor Spo0A, presumably acting as a signaling cofactor with histidine kinase KinD and protein components of phosphorele, Spo0F and Spo0B phosphotransferases [28]. The effect of manganese ion stimulation on biofilm formation has been found for many bacilli and suggests that this signaling pathway for biofilm formation is highly conserved for this group of microorganisms [29,30].

The effect of Mg^2+^ ions at final concentrations of 1, 3, and 9 mM on the development of *B. subtilis* 168 biofilm was studied (Figure 2). It was established that Mg^2+^ ions at a concentration of 1–3 mM contribute to biofilm structure at the early stages of its organization; after 24 h, we observed a decrease in film formation of up to 10% in the presence of 3 mM Mg^2+^. An increase in the magnesium concentration in the medium to 9 mM led to a decrease in the level of biofilms of *B. subtilis* 168 by 60%. Per the literature data, Mg^2+^ ions play an important role in regulating the adhesion of cells that have lost their mobility [31]. The authors emphasize that the additional introduction of divalent magnesium cations into the cultivation solution leads to a significant increase in adhesion during biofilm colonization of bacilli.

Zn^2+^ ions at concentrations of 1 and 3 mM did not affect the formation of *B. subtilis* 168 biofilms within 12 h of growth. However, at a Zn^2+^ concentration of 9 mM, a decrease in the level of biofilm formation (from 60% to 70%) was observed up to 48 h (Figure 2). Moreover, even at high concentrations of Zn^2+^ ions, only partial suppression of biofilm formation was observed under structured community conditions. The toxic effect of Zn^2+^ ions is associated with its ability to complex formation, which in turn can disrupt protein function in the cell membrane including enzyme systems of the respiratory chain [32].

The addition of 1 M NaCl to the culture medium led to a 1.5-fold decrease in the formation of *B. subtilis* 168 biofilms by the 48th hour of growth (Figure 2c). In the presence of 1 M NaCl, a structured community of halotolerant bacteria is preserved in the biofilm, which supports population survival under salt stress. There is evidence that *B. subtilis* histidine kinase can function as an osmosensor due to the extracellular domain of CACHE, which seems to be important for the continued colonization of *B. subtilis* under stress conditions [28].

Reactive oxygen species and oxidizing agents, such as H_2_O_2_, are involved in lipid peroxidation, protein degradation, and enzyme inactivation, causing significant damage to bacteria [33]. Bacteria have developed protective mechanisms against oxidative stress. Among such mechanisms are the synthesis of catalase and superoxide dismutase, essential for the removal of free radicals. Three families of catalase genes have been identified in the genomes of bacteria, namely: monofunctional catalases, heme-containing bifunctional catalase-peroxidases, and Mn-containing catalases [33,34]. Together they provide a well-balanced defense system for aerobic bacteria to ensure the survival of living cells by removing reactive oxygen species.

The effect of oxidative stress on the formation of a *B. subtilis* 168 biofilm was studied when hydrogen peroxide was added to the cultivation medium at concentrations from 1 to 10 mM. Inhibition of the growth of planktonic culture by 1.9 times for 48 h was observed. At a peroxide concentration of 10 mM (Figure 3A), 1.4-fold suppression of biofilm development was registered under the same conditions (Figure 3B). In addition, the degree of inhibition increased with increasing peroxide concentration; however, the degree of resistance to peroxide in a biofilm culture is higher than that of planktonic culture. Thus, H_2_O_2_ inhibits film formation to a lesser extent than growth in free bacteria.

To comparatively study the formation of *B. subtilis* biofilm, we used regulatory mutants, i.e., strains whose genome contains inactivated genes for global transcription factors that affect the formation of bacilli biofilms: Spo0A, DegU, and AbrB [5,6,7]. The following strains were used in the experiment: *B. subtilis* 168 (wild type), *B. subtilis* abrB^−^ with an inactivated abrB gene (pleiotropic regulator of the vegetative phase), *B. subtilis* degU^−^ with an inactivated degU gene (a component of the DegS/DegU regulation system), and *B. subtilis* spo0A^−^—with an inactivated *spo0A* gene (stationary phase regulator). The formation of biofilms by these strains was studied (Figure 4).

The strain with the inactivated pleiotropic repressor AbrB showed an increase in the level of biofilm formation of up to 20% within the first 48 h; further, the level of film formation remained similar to that of the control (Figure 4). The *degU*-mutant showed a decrease in biofilm formation over 72 h, with a maximum decrease of up to 40% at 48 h. *spo0A*-deficiency contributed to a decrease in the level of film formation for the first 48 h of growth, while at the late stage (after 60 h) in *spo0A* mutants, an increase in the level of biofilm formation was observed in comparison to the wild-type strain (Figure 4). At later stages, the transcription factor Spo0A initiates the sporulation process. Our data are consistent with genetic data that, at the stage of vegetative growth, the AbrB repressor is involved in the suppression of the expression of operon genes encoding the biofilm matrix; later, the DegU regulator is involved in the activation of biofilm matrix genes; at the biofilm dispersion stage, the Spo0A factor blocks the expression of biofilm matrix genes and activates spore formation. The process of biofilm formation is complex and its regulation is under the control of a multiple-branched regulatory network. In our experiments, none of the regulatory proteins blocked the formation of biofilm completely. This process is subject to the integrated control of regulatory systems that evaluate the input of many different signals. Therefore, the inactivation of one of the regulators was not critical; in the absence of one of the regulatory proteins, the biofilm is formed under the control of another regulatory network.

The biofilms of the *spo0A*-mutant and the wild-type strain, showed maximum growth at 37 °C and pH 7.4 (Figure 5a). However, the response of the *spo0A*-mutant to the presence of metals differed from that of the wild-type strain: in the presence of 3 mM Ca^2+^ and 3 mM Mn^2+^ ions in the medium, biofilm inhibition was observed at a level of up to 20%; this effect was absent for Mg^2+^ at concentrations of 3 mM. The data reflect the participation of Ca^2+^ and Mn^2+^ in activating the phosphorylation of the transcription factor Spo0A (Figure 5a). The toxic effect of Zn^2+^ on film formation was only noticed in the wild-type strain. Data on the resistance of the *spo0A*-mutant to 1 M NaCl are comparable with that of the wild-type strain.

Activation of the Spo0A factor depends on the phosphorylating action of at least five different kinases (KinA–E), which are responsible for the transfer of the phosphate group to the Spo0A regulator [35,36]. Kinase activation is regulated by specific signals, some of which have been identified. KinC induces low levels of Spo0A~P activation in response to changes in the content of potassium cations [37]. KinD is sensitive to the content of manganese ions [28,38]. KinB is activated due to interruptions in electron transport resulting from environmental influences (low oxygen content or large concentrations of iron ions). KinA responds to NAD/NADH levels in the cytoplasm [39]. The level of Spo0A activation is determined by the cumulative signal input, in which metal ions are involved as cofactors. The obtained data on an increase in the level of film formation of *spo0A* mutants in the presence of Ca^2+^ and Mn^2+^ ions indicate an important role of these metals in the activation of the Spo0A transcription factor. The activated form of Spo0A~P in turn inhibits the expression of AbrB, which is an alternative repressor for the expression of biofilm matrix genes [37].

Interesting results were obtained for the *degU^−^* mutant biofilm (Figure 6). The optimal conditions for biofilm formation were 37 °C and pH 7.4, and were the same as the optimal conditions for the wild-type strain. However, the dynamics of biofilm formation changed with the presence of divalent metal ions: the biofilm reached its maximum level at 36 h of growth, followed by the stage of biofilm dispersion (Figure 6). In contrast to the *spo0A*-mutant, biofilm formation was activated in the presence of 1–3 mM Ca^2+^, Mg^2+^, and Mn^2+^ averaged by ~20%, with a maximum effect at a concentration of 1 mM. With an increase in concentration to 9 mM, the film formation activity decreased with a profound effect by Mg^2+^ and Mn^2+^ ions. For Zn^2+^ ions, the resistance of the *degU*-mutant biofilm to their toxic effects increased by 10% compared to the wild-type strain. When 1 M sodium chloride was introduced into the medium, the ability to form a biofilm in the *deg*-mutant decreased by 70% within 48 h, as compared to 30% for the cells of the wild-type strain under identical conditions. Thus, the inactivation of the transcription factor DegU leads to an early dispersion of the biofilm and a decrease in the strain’s salt tolerance.

The *abrB*-mutant formed biofilms at the same optimal temperature and pH as the wild strain (Figure 7a,b). The mutation did not affect the level of biofilm formation when 1 M sodium chloride was added to the medium (Figure 7c). In the presence of Ca^2+^, Mg^2+^, and Mn^2+^ at concentrations ≤ 3 mM, biofilm formation was activated with the maximum effect (by 17%) of 1 mM Mn^2+^. Increasing the concentration to 9 mM declined the film-forming activity, with a greater effect by Mg^2+^ and Mn^2+^ ions. The resistance of the *abrB*-mutant biofilm to the toxic effect of Zn^2+^ ions increased by 10% compared to the wild-type strain. A similar effect was also observed in the *deg*-mutant. Thus, biofilm formation of the *abrB*-mutant depends on the presence of metal ions in the medium with a strong effect by 3 mM Mn^2+^ ions.

Biofilms of *B. subtilis* are the main type of cooperative existence of bacteria in the rhizosphere. In addition to physical and chemical factors, bacteria united by a biofilm are regulated and differentiated into subpopulations inside the biofilm under enzymatic influences. There is a dearth of information on the effect of enzymes on biofilm development. Such knowledge is essential for the development of a general strategy for the management of biofilms of spore-forming *B. subtilis* bacteria, which are considered to be an effective biocontrol tool.

We studied the contribution of the proteolytic activity of bacilli to the formation of *B. subtilis* biofilm; these bacteria effectively secrete various proteases into the medium. *B. subtilis* strains defective in the synthesis of extracellular proteases were used. 62 protease genes have been identified in the *B. subtilis* genome, of which 14 are secreted [40]. We used the *B. subtilis* 20–36 strain, with two inactivated extracellular proteinases, the *B. subtilis* BRB8 strain with nine inactivated extracellular proteinases, and the *B. subtilis* BRB14 strain, with eleven inactivated extracellular proteinases. The dynamics of biofilm formation by the mutant strains were studied at pH 7.4 and a cultivation temperature of 37 °C (Figure 8).

The mutant strains demonstrated an increase in the level of biofilm formation compared to the wild-type strain, and the greater the number of inactivated proteinase genes, the higher the level of biofilm formation (Figure 8). The maximum difference in biofilm optical density from the wild-type strain was 15%, which was registered in the mutant strain with 11 inactivated genes of extracellular proteinases (Figure 8).

A comparative study of the characteristics of the biofilm of the *B. subtilis* BRB14 strain showed that the mutant strain formed a biofilm analogous to that of the wild-type while maintaining a film-formation maximum at 48 h (Figure 9a,b). The biofilm of *B. subtilis* BRB14 was more resistant to 1 M NaCl relative to that of the wild-type strain (Figure 9c). In the presence of Ca^2+^, Mg^2+^, and Mn^2+^ ions in the medium at concentrations ≤ 3 mM, biofilm formation was sustainably activated with a maximum effect by Mn^2+^ ions at concentrations ≤ 3 mM (up to 10%). Ca^2+^ and Mg^2+^ concentrations ≤ 3 mM showed a positive effect just until the 36th h. With an upsurge in concentration to 9 mM, the film formation activity declined, but to a lesser extent as compared to the wild-type strain. The resistance of the *B. subtilis* BRB14 biofilm to the toxic effect of Zn^2+^ ions increased by 6% relative to the wild-type strain. Comparative analysis depicted the *B. subtilis* BRB14 biofilm as more resistant to 1 M NaCl than that of the other mutant strains.

Extracellular proteases can potentially participate in the processes of control over the formation of the biofilm matrix during the processing of regulatory peptides, as well as in the dispersion of the biofilm. All genetic cascades of *B. subtilis* are negatively controlled by the Rap family of phosphatases that specifically dephosphorylate global regulators. Each Rap phosphatase is inactivated by a specific cognate extracellular regulatory peptide (Phr peptides) [41]. Rap-Phr pairs coordinate various biological processes with population density, including biofilms [42]. RapGH phosphatase is known to dephosphorylate DegU~P while RapABEJ phosphatases dephosphorylate Spo0A~P. Regulatory peptides are an alternative signaling program between cells as a mode of interaction [43]. Extracellular proteinases may be involved in peptide turnover [40]. Our data showed that the inactivation of 11 genes of *B. subtilis* extracellular proteinases leads to an increase in film formation (Figure 8).

A common feature of bacilli biofilms is the abundance of architectural features, the complexity of which depends on ecological and stress factors of the environment and is controlled by the cooperative interaction of regulatory networks. The ultrastructure of *B. subtilis* 168 biofilms and mutant strains defective in transcription factor genes and extracellular proteinase genes (BRB14) was studied by scanning electron microscopy (Figure 10 168). The *B. subtilis* BRB14 biofilm is distinguished by a more intensive formation of an extracellular matrix with a branched structure (Figure 10 BRB14). The absence of extracellular proteinases leads to the formation of a biofilm with a denser structure. The biofilm of the *spo0A*-mutant is characterized by the formation of a matrix with embedded bacilli cells, while the three-dimensional structure of the *degU*-mutant biofilm differs in morphology and more pronounced architecture compared to the biofilm of the *spo0A*-mutant and the wild type strain (Figure 10 *spo0A^−^*, *degU^−^*). The biofilm of the *abrB*-mutant differs from the others and contains more matrix; the cells are immersed in it, which is associated with the inactivation of the repressor of the matrix gene operons (Figure 10 *abrB^−^*). Inactivation of each of these genes has a different effect on the structure and morphology of the biofilm matrix as a result of changes in regulatory networks; however, even if one of the transcription factors is switched off, bacilli proceed to form biofilms, which confirms the complexity of the ways to control its formation.

The three-dimensional architecture of the *B. subtilis* biofilm has been understudied. The extracellular biofilm matrix of *B. subtilis* includes an exopolysaccharide that is critical for community architecture and function. Understanding of the mechanism of biosynthesis and molecular composition of *B. subtilis* exopolysaccharide remains incomplete [44]. The structural integrity of the matrix is provided by two secreted proteins, TasA and TapA, which are encoded by the operon of three *tapA-sipW-tasA* genes (*tapA* operon) [45]. TasA is a functional amyloid protein that is secreted into the extracellular space, where it self-assembles into fibers that are attached to the cell wall by the TapA protein [46]. This architecture ensures the resistance of the biofilm to external influences.

The ability to synthesize amyloid-like proteins in the biofilm of *B. subtilis* strains was established by testing on medium E with the Congo Red dye (Figure 11).

On the 2nd day of cultivation, the colonies on the agar medium turned red, indicating the presence of amyloid-like proteins in the *B. subtilis* biofilm. We studied the binding of the Congo Red dye by extracellular amyloids of biofilms of different strains of *B. subtilis*: a protease-deficient strain and regulatory mutants *spo0A^−^* and *degU^−^* and compared with the data for the wild-type strain *B. subtilis* 168 (Figure 11). It was found that relative to the wild-type strain, BRB14 and the degU mutant contain more amyloid-like proteins in the biofilm, while the spo0A mutant, on the contrary, contains less. Based on the results of studies on the ultrastructure and relative content of amyloid-like proteins in biofilms of different strains, it can be concluded that extracellular proteinases can participate in the dissociation of the protein components of the *B. subtilis* biofilm and contribute to the functioning of this complex structure at the dispersion stage.

## 4. Conclusions

Biofilms and sporulation are predominantly the natural modes of bacilli life, in particular in the rhizosphere. Biofilm formation is a flexible system of adaptation to environmental factors and a symbiotic way of interacting with macroorganisms such as plants, for example. The processes of biofilm formation and sporulation are regulatorily separated and do not occur simultaneously but develop in accordance with incoming signals. Biofilms of *B. subtilis* develop at a temperature range of 20 °C to 45 °C and a pH range of 6.5–8.5; sporulation begins outside these parameters. The *B. subtilis* genome contains genes regulating mechanisms of biofilm formation and development. These mechanisms are based on complex genetic programs that are implemented in compliance with a combination of external factors and signals. A study of the ultrastructure of biofilms of *B. subtilis* mutant strains defective in the genes of extracellular proteinases and regulatory proteins showed that they differ in architecture, structural heterogeneity of the matrix, and in the content of amyloid-like proteins, the maximum level of which was established for *deg*-mutants and strains defective in extracellular proteinases. The level of biofilm formation is partially inhibited in *deg*-mutants by about 30–40%, and upsurged in *abrB*-mutants by 25% relative to the wild-type strain. This means that the loss of one of the regulators Spo0A, DegU, and AbrB did not cause a critical change in biofilm formation; together they exert multiple control of film formation. Since *B. subtilis* naturally inhabits more complex environments, such as plant roots, the presence of other bacteria, and rhizosphere organisms, control mechanisms, and interactions can even be more bewildering.

## Figures and Tables

**Figure 1 microorganisms-11-01486-f001:**
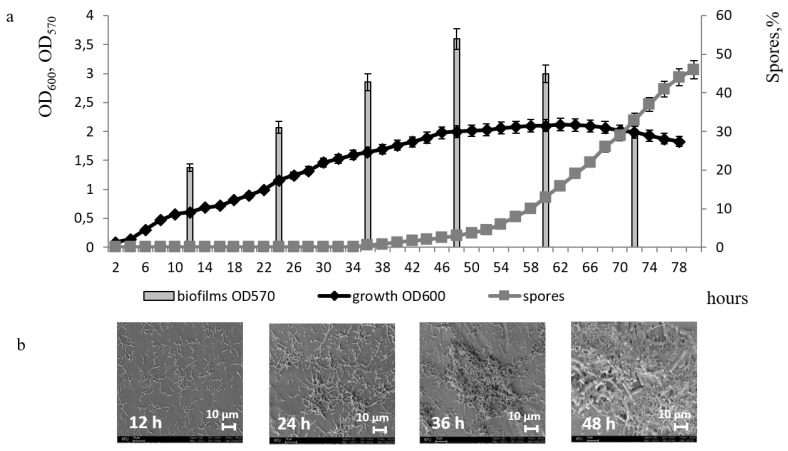
Dynamics of growth, sporulation, and biofilm formation of *B. subtilis* 168 (**a**); Scanning electron microscopy (SEM) of the biofilm surface of *B. subtilis* 168; ×1000 k (**b**).

**Figure 2 microorganisms-11-01486-f002:**
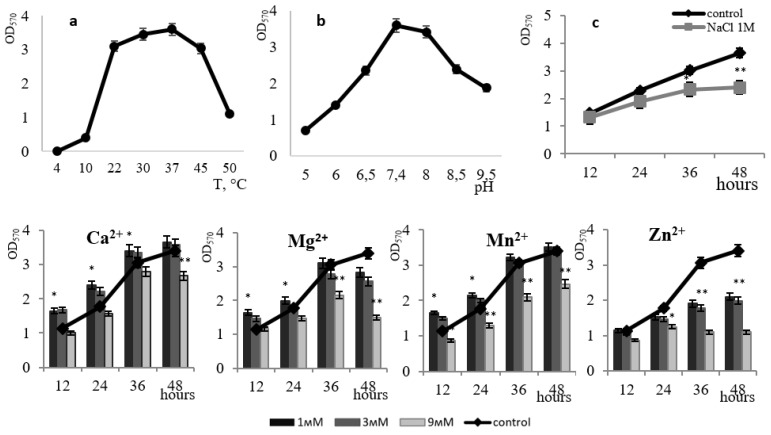
The effect of temperature (**a**) and pH (**b**) on biofilm formation by the *B. subtilis* 168 strain. The effect of 1 M NaCl on the growth of *B. subtilis* 168 biofilm at pH 7.4 and temperature 37 °C (**c**). The effect of divalent metal ions (Ca^2+^, Mg^2+^, Mn^2+^, Zn^2+^) on *B. subtilis* 168 biofilm formation. * *p* ≤ 0.05; ** *p* ≤ 0.01.

**Figure 3 microorganisms-11-01486-f003:**
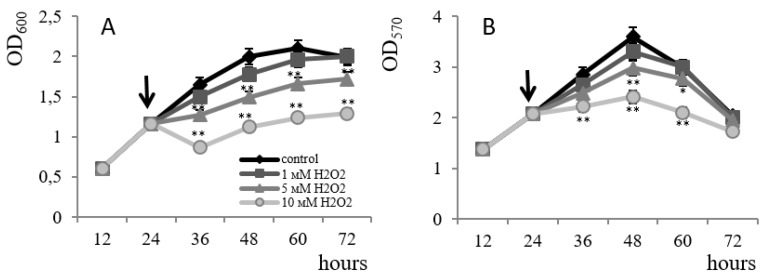
The effect of hydrogen peroxide on the growth of plankton culture (**A**) and the formation of biofilms (**B**) of strain *B. subtilis* 168. The arrow indicates the time of hydrogen peroxide introduction into the culture. * *p* ≤ 0.05; ** *p* ≤ 0.01.

**Figure 4 microorganisms-11-01486-f004:**
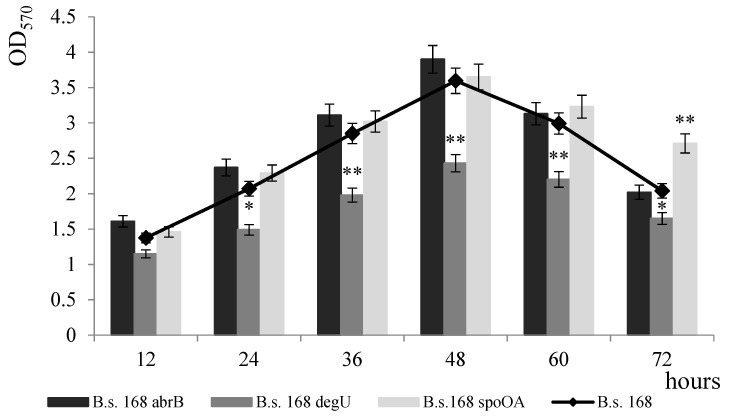
Formation of biofilms by *B. subtilis* strains with inactivated genes of regulatory proteins (*B. subtilis* abrB^−^, *B. subtilis* degU^−^, *B. subtilis* spo0A^−^). * *p* ≤ 0.05; ** *p* ≤ 0.01.

**Figure 5 microorganisms-11-01486-f005:**
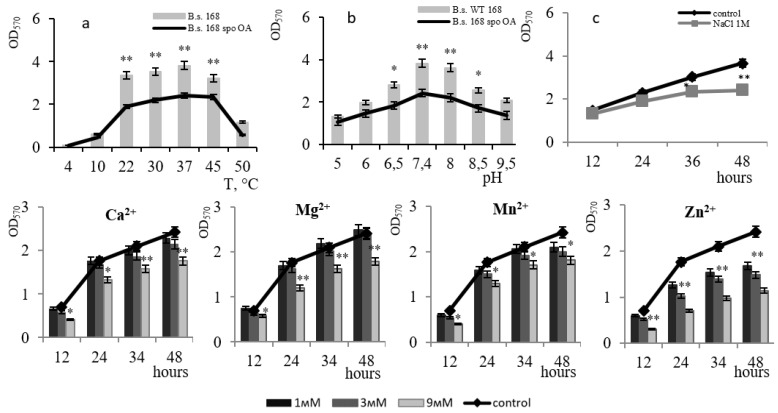
The effect of temperature (**a**) and pH (**b**) on biofilm formation by *B. subtilis* spo0A^−^. The effect of 1 M NaCl on *B. subtilis* spo0A biofilm growth at pH 7.4 and 37 °C (**c**). The effect of divalent metal ions (Ca^2+^, Mg^2+^, Mn^2+^, Zn^2+^) on biofilm formation by *B. subtilis* spo0A. * *p* ≤ 0.05; ** *p* ≤ 0.01.

**Figure 6 microorganisms-11-01486-f006:**
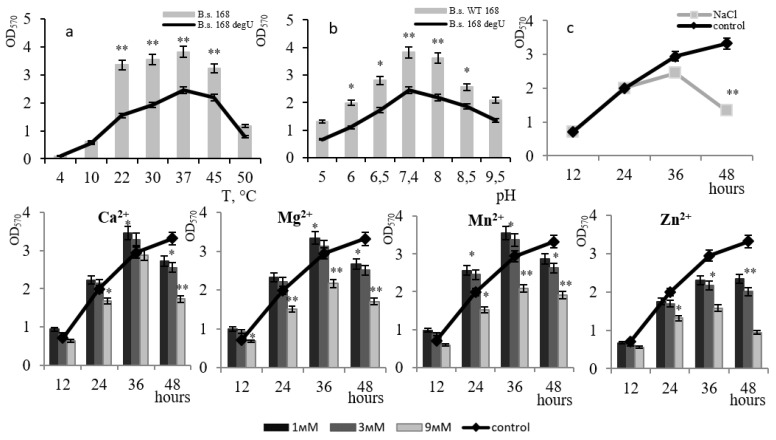
The effect of temperature (**a**) and pH (**b**) on biofilm formation by *B. subtilis* degU^−^. The effect of 1 M NaCl on biofilm development of the *B. subtilis* degU^−^ strain at pH 7.4 and 37 °C (**c**). The effect of divalent metal ions (Ca^2+^, Mg^2+^, Mn^2+^, Zn^2+^) on biofilm formation by *B. subtilis* degU^−^ strain. * *p* ≤ 0.05; ***p* ≤ 0.01.

**Figure 7 microorganisms-11-01486-f007:**
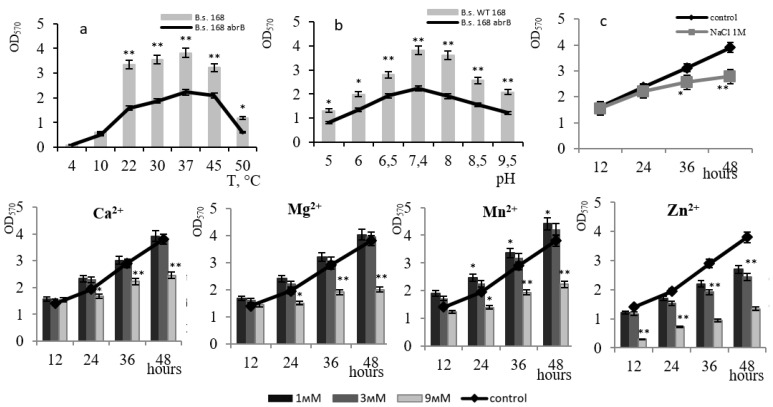
The effect of temperature (**a**) and pH (**b**) on biofilm formation by *B. subtilis* abrB^−^ strain. The effect of 1 M NaCl on the biofilm formation of the *B. subtilis* abrB^−^ strain at pH 7.4 and 37 °C (**c**). The effect of divalent metal ions (Ca^2+^, Mg^2+^, Mn^2+^, Zn^2+^) on biofilm formation by *B. subtilis* abrB^−^ strain. * *p* ≤ 0.05; ** *p* ≤ 0.01.

**Figure 8 microorganisms-11-01486-f008:**
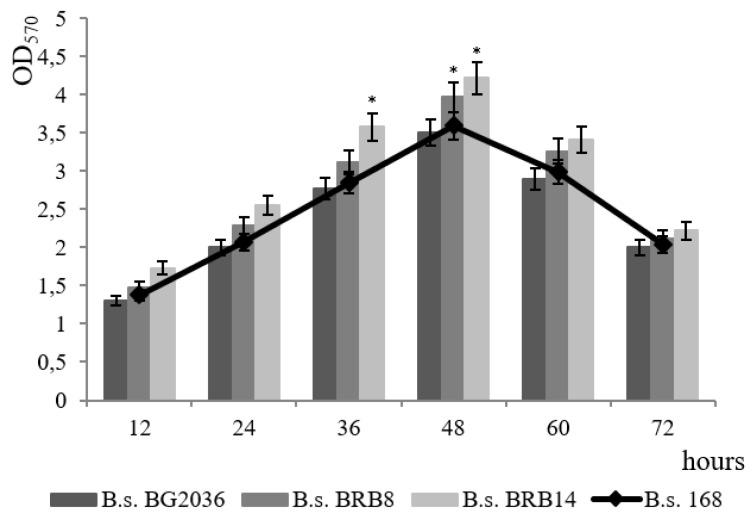
Biofilm formation by protease-deficient strains *B. subtilis* BG2036, *B. subtilis* BRB8 and *B. subtilis* BRB14. The *B. subtilis* 168 strain was taken as a control for comparison. * *p* ≤ 0.05.

**Figure 9 microorganisms-11-01486-f009:**
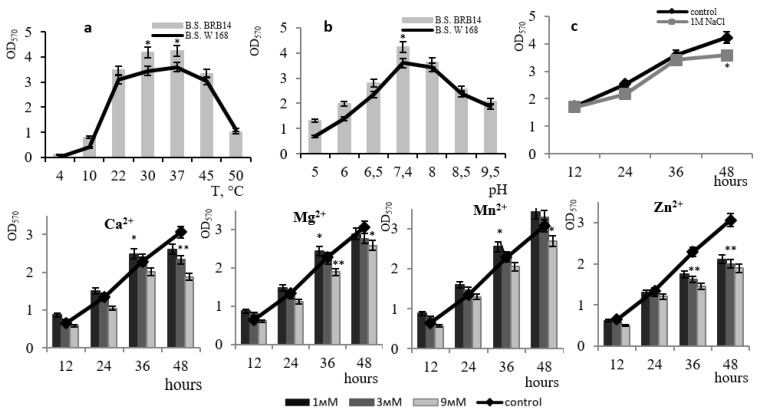
The effect of temperature (**a**) and pH (**b**) on biofilm formation by *B. subtilis* BRB14. The effect of 1 M NaCl on the growth of *B. subtilis* BRB14 biofilm at pH 7.4 and 37 °C (**c**). The effect of divalent metal ions (Ca^2+^, Mg^2+^, Mn^2+^, Zn^2+^) on the formation of *B. subtilis* BRB14 biofilm. * *p* ≤ 0.05; ** *p* ≤ 0.01.

**Figure 10 microorganisms-11-01486-f010:**
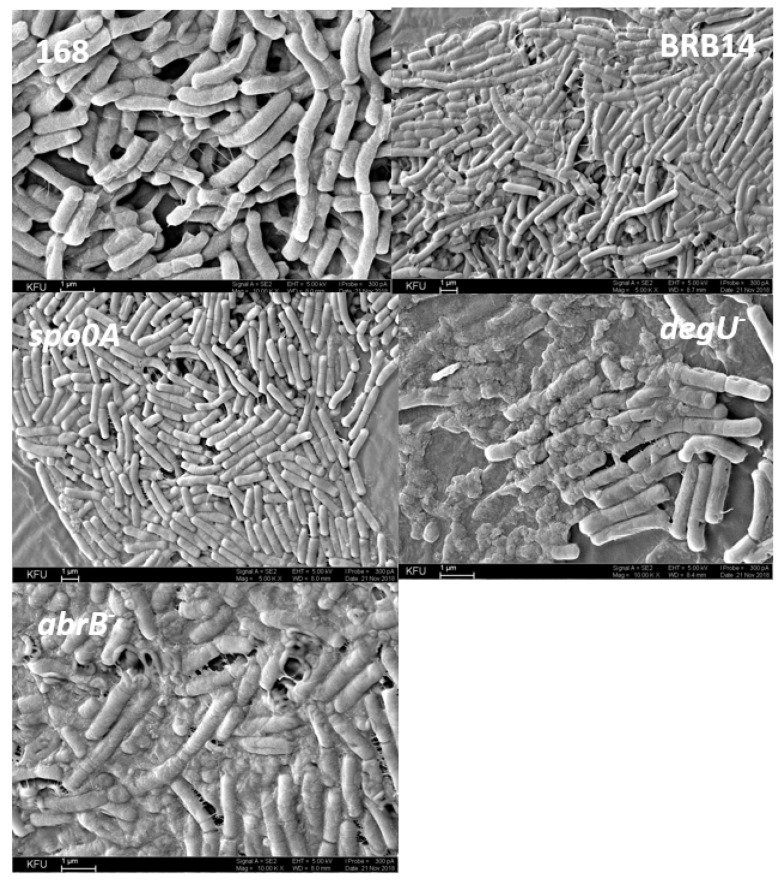
Scanning electron microscopy of the surface of a 48-h biofilm of *B. subtilis* 168, *B. subtilis* BRB14, *B. subtilis* spo0A^−^, *B. subtilis* degU^−^, *B. subtilis* abrB^−^; ×1000 k, mark 1 µm.

**Figure 11 microorganisms-11-01486-f011:**
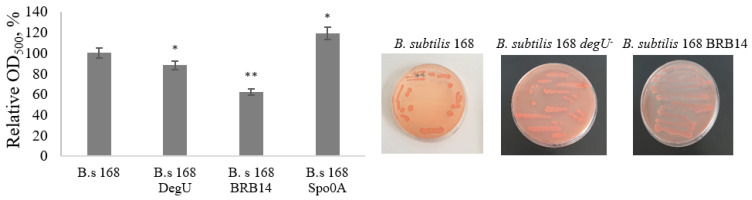
Binding of Congo Red dye to extracellular amyloids of mutant strains and the wild-type strain of *B. subtilis* at 48 h of biofilm formation. * *p* ≤ 0.05; ** *p* ≤ 0.01.

**Table 1 microorganisms-11-01486-t001:** Strains used in the study.

Strains	Mutation Description	Source
*Bacillus subtilis* WT 168	Natural isolate (wild-type)	Proffessor J. Stuelke, University of Göttingen, Germany
Protease-deficient strains		
*Bacillus subtilis* BG20-36	ΔnprE–522; Δapr–684	Eugenio Ferrarri, Genencor Int. Inc., Rochester, NY, USA
*Bacillus subtilis* BRB08	ΔtrpC2, ΔnprB, ΔaprE, Δepr, Δbpr, ΔnprE, Δmpr, Δvpr, ΔwprA	Cobra Biologics, Keele, UK
*Bacillus subtilis* BRB14	ΔtrpC2, ΔnprB, ΔaprE, Δepr, Δbpr, ΔnprE, Δmpr, Δvpr, ΔwprA, ΔhtrA, ΔhtrB	Cobra Biologics, Keele, UK
Strains with regulatory mutations		
*Bacillus subtilis* 168 abrB::kan (amyE::pAT606)	ΔabrB (Kan)	Dr. Prof. T. Masher Ludwig-Maximillian’s University, Munich, Germany
*Bacillus subtilis* 168 degU::kan (amyE::pAT606)	ΔdegU (Kan)	Dr. Prof. T. Masher Ludwig-Maximillian’s University, Munich, Germany
*Bacillus subtilis* 168 spo0A::tet (amyE::pAT612)	Δspo0A (Tet)	Dr. Prof. T. Masher Ludwig-Maximillian’s University, Munich, Germany

## Data Availability

The data obtained in this study are available upon request.

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
