# Peer review of "Biofilm Formation by Mutant Strains of Bacilli under Different Stress Conditions"

_microorganisms, 2023, doi:10.3390/microorganisms11061486_

Round 1

Reviewer 1 Report

Abstract

Needs all parts. I mean: introduction, objective, methodology and results.

Key words: do not use words in title as key words

Introduction

Lines  65-and 66 “a synthetic-E medium, whose composi- 65 tion is described in [13]” needs at least aa short description

Lines 78-70 “Biofilm formation defined by the method set incubation with crys- 78 tal violet (CV) [14] with modification [15]” needs at least aa short description.

Methodology

Experimental set up?  Experimental design? Temperature, ph, metals and H2O2 are the independent variables.

General comments

In general methodology lack of detailed information. Methodology must provide all detailed information to replicate the study.

Consider to change the title. Through the MS the factors that induce stress are not described clearly. On my opinion the title does not describe the content on the MS.

See some notes in MS attached

Minor revision needed

Reviewer 2 Report

The authors have carried out labor-intensive experimental work on the influence of biological and genetic factors on the formation of microbial biofilms, using the example of soil strains of B. subtilis, at a high professional level. Each such work is relevant, promising and valuable for the scientific community, since the mechanism of biofilm formation is very complex and confusing, and the experiments conducted require careful planning to obtain conclusive results.

I believe that the authors have fulfilled their objectives and achieved the stated goal of the study. The work is complete, the text is well structured, the figures are sufficiently clear and understandable.

Questions and suggestions for improving the work:

1) What instrument was used to determine optical density?

2) A brief description of modification of biofilm formation assay and what was used as a solvent for solubilization would be helpful?

3) In the text indicate the number of repetitions of experiments and what was used as a control.

4) It is not clear from which museum collections B. subtilis strains were obtained or if these strains have official registration?

Round 2

Reviewer 1 Report

Thank you for the corrections.